# Antimicrobial Peptide L18R Displays a Modulating Action against Inter-Kingdom Biofilms in the Lubbock Chronic Wound Biofilm Model

**DOI:** 10.3390/microorganisms9081779

**Published:** 2021-08-21

**Authors:** Paola Di Fermo, Tecla Ciociola, Silvia Di Lodovico, Simonetta D’Ercole, Morena Petrini, Laura Giovati, Stefania Conti, Mara Di Giulio, Luigina Cellini

**Affiliations:** 1Department of Pharmacy, University “G. d’Annunzio” Chieti-Pescara, 66100 Chieti, Italy; paola.difermo@unich.it (P.D.F.); silvia.dilodovico@unich.it (S.D.L.); l.cellini@unich.it (L.C.); 2Department of Medicine and Surgery, University of Parma, 43125 Parma, Italy; tecla.ciociola@unipr.it (T.C.); laura.giovati@unipr.it (L.G.); stefania.conti@unipr.it (S.C.); 3Department of Medical, Oral and Biotechnological Sciences, University of “G. d’Annunzio” Chieti-Pescara, 66100 Chieti, Italy; simonetta.dercole@unich.it (S.D.); morena.petrini@unich.it (M.P.)

**Keywords:** antimicrobial peptide, chronic wounds, *Staphylococcus aureus*, *Pseudomonas aeruginosa*, *Candida albicans*, inter-kingdom biofilm, Lubbock chronic wound biofilm model, confocal laser scanning microscopy, scanning electron microscopy

## Abstract

Chronic wound infections represent an important health problem due to the reduced response to antimicrobial treatment of the pathogens organized in structured biofilms. This study investigated the effects of the previously described antifungal peptide L18R against three representative wound pathogens: *Staphylococcus aureus, Pseudomonas aeruginosa*, and *Candida albicans.* The antimicrobial activity of L18R was evaluated (i) against single planktonic microbial populations; (ii) on single, dual, and triadic species of biofilms in both the early stage and mature stage; and (iii) in the polymicrobial Lubbock chronic wound biofilm (LCWB) model, mimicking spatial microbial colonization. This study used the evaluation of CFUs, biofilm biomass detection, and confocal and scanning electron microscopy analysis. L18R showed a significant antimicrobial activity against planktonic microorganisms and was able to differentially reduce the biomass of monomicrobial biofilms. No reduction of biomass was observed against the polymicrobial biofilm. In mature LCWB, L18R caused a moderate reduction in total CFU number, with a variable effect on the different microorganisms. Microscopy images confirmed a predominant presence of *P.*
*aeruginosa* and a lower percentage of *C. albicans* cells. These findings suggest a modulating action of L18R and recommend further studies on its potential role in chronic wound management in association with conventional antibiotics or alternative treatments.

## 1. Introduction

Chronic ulcers are known to be a major global health problem and have become a challenge to healthcare systems worldwide [1]. Chronic wounds do not progress through the healing process in a timely manner, they last on average 12 to 13 months, and recur in up to 60–70% of patients with vascular disease, diabetic ulcers, pressure ulcers, and other comorbid conditions [2]. These ulcers usually lead to a worsening of quality of life, becoming more difficult to treat and often associated with high treatment costs [3].

The delayed healing process of chronic wounds is a result of pathophysiologic phenomena including prolonged or increased inflammation, inability of dermal and/or epidermal cells to respond to reparative stimuli, persistent infections, and formation of multispecies drug-resistant microbial biofilms [4]. Polymicrobial infections that occur in chronic wounds mainly involve microorganisms such as *Pseudomonas aeruginosa* and *Staphylococcus aureus*, which are the two most commonly co-isolated microbial species; a defined spatial distribution has been reported, where, generally, *S. aureus* is located near the wound surface while *P. aeruginosa* colonizes the deeper area of the chronic wound bed [5].

The role of yeasts in wound biofilms is under-recognized and under-appreciated, although reports show an important contribution in the composition of wound microbiomes. *Candida* spp. are most frequently isolated; in particular, *C. albicans* is found in polymicrobial inter-kingdom yeast-bacterial biofilms [6]. The multispecies nature of chronic wound microbiota and the capability of the associated microorganisms to interact/cooperate lead to resilient responses against traditional antibacterial and antifungal treatments.

Although wound biofilms are characterized by a polymicrobial composition, several studies have only taken into account single-species biofilms for the evaluation of the effects of innovative/natural/alternative therapeutic approaches, underestimating the polymicrobial response to treatments [7,8,9,10].

The Lubbock chronic wound biofilm (LCWB) model represents a valid in vitro chronic wound system for multispecies consortium studies [11]. In this model, red blood cells, plasma, and nutrients closely mimic the wound bed condition and easily allow for the biofilm growth of a multispecies microbial population. Moreover, *S. aureus* coagulase activity produces a fibrin network that designs and arranges a wound-like biofilm structure, creating a scaffold for microbial adhesion.

Over the last few years, antimicrobial peptides (AMPs) of various origins have attracted great attention as potential anti-infective agents to face the spreading resistance to conventional antibiotics; AMPs can lead bacteria to develop no or low resistance [12,13,14,15].

This study focused on the evaluation of the antibiofilm activity of a previously described fungicidal peptide. The selected molecule, L18R, is the translated product of immunoglobulin gene J (locus heavy, IGHJ2) that showed fungicidal activity at micromolar concentrations against reference yeast strains, including *Candida* strains resistant to conventional antifungal drugs [16]. In addition, L18R proved to be effective in an experimental model of systemic infection by *C. albicans* in larvae of *Galleria mellonella*, displaying no hemolytic, cytotoxic, or genotoxic activity in mammalian cells [16].

In the first step of this study, the antimicrobial activity of L18R was evaluated against the planktonic form of three clinical isolates representative of wound pathogens: *S. aureus, P. aeruginosa*, and *C. albicans*. Subsequently, the peptide effects on single, dual, and triadic species biofilms were analyzed both in the early stages of biofilm formation and in mature biofilms. Moreover, the effect of L18R treatment was also studied when the above-mentioned microorganisms were grown in the LCWB model to better mimic the physiological environment in chronic wounds.

## 2. Materials and Methods

### 2.1. Microbial Cultures

Clinical strains from anonymized patients of *S. aureus* PECHA 10, *P. aeruginosa* PECHA 4, and *C. albicans* X3 derived from patients with chronic wounds [17] were used; all patients gave their informed consent for the study. Clinical strains were isolated from chronic wound swab culturings onto Mannitol salt agar (MSA, Oxoid, Milan, Italy), Cetrimide agar (CET, Oxoid, Milan, Italy), or Sabouraud agar medium (SAB, Oxoid, Milan, Italy). All plates, except SAB plates, were incubated aerobically at 37 °C for 24 h. SAB plates were incubated aerobically at 30 °C for 48 h. Microorganisms were identified by morphologic aspects of the colonies, followed by biochemical identification using the automated Vitek 2 system (bioMerieux, Marcy l’Etoile, France. The study (ID n. richycnvw) was approved by the Inter Institutional Ethics Committee of University “G. d’Annunzio” Chieti-Pescara, Chieti, Italy. For the experiments, these microorganisms (*S. aureus* PECHA 10, *P. aeruginosa* PECHA 4, and *C. albicans* X3) came from the private collection of the Bacteriological Laboratory of the Pharmacy Department, University “G. d’Annunzio” Chieti-Pescara and were cultured on MSA, CET, and SAB, respectively, for 24 h at 37 °C.

### 2.2. Evaluation of L18R Activity against Planktonic Bacterial and Fungal Cells

L18R peptide (amino acid sequence: LLVLRSLGPWHPGHCLLR) was synthesized, purified, and solubilized as previously described [16]. Preliminarily, the activity of L18R was evaluated in vitro against planktonic bacterial and yeast cells by previously described CFU assays [16,18]. Briefly, bacterial and fungal suspensions (nearly 500 CFUs) were incubated in absence (control) or presence of serial concentrations of L18R at 37 °C for 5 and 6 h, then plated on Muller Hinton agar and SAB, respectively. CFUs were enumerated after 24–48 h of incubation at 37 °C and peptide fungicidal activity was determined as a percentage of CFU inhibition. Each assay was carried out in triplicate and at least two independent experiments were performed for each condition. EC_50_ was calculated by nonlinear regression analysis using GraphPad Prism 5 software.

### 2.3. Biofilm Assays

For biofilm formation assays, bacteria were cultured in Trypticase soy broth (TSB, Oxoid, Milan, Italy) and incubated at 37 °C overnight in aerobic conditions and then standardized in TSB with 0.5% glucose (TSBG) at an optical density (OD_600_) = 0.15 for *S. aureus* PECHA 10 and 0.30 for *P. aeruginosa* PECHA 4. *Candida albicans* X3 was recovered by SAB plates and the suspension standardized at a final concentration of 1 × 10^6^ cells/mL in RPMI-1640 medium (Sigma-Aldrich, Milan, Italy) with 2% glucose (RPMI-G) for the monomicrobial biofilms and in TSBG for the polymicrobial biofilms.

The effect of the treatment with L18R was evaluated both on the early-stage biofilms and on the mature biofilms. Preliminary assays were carried out against monomicrobial biofilms, then against polymicrobial biofilms containing *S. aureus* and *P. aeruginosa* (MIX 2) or *S. aureus*, *P. aeruginosa,* and *C. albicans* (MIX 3).

#### 2.3.1. L18R Treatment on the Early Stage of Biofilm Formation

For studies on monomicrobial biofilm, 100 μL of standardized *S. aureus* PECHA 10, *P. aeruginosa* PECHA 4, or *C. albicans* X3 were dispensed into 96-well polystyrene flat-bottomed microtiter plates (3 each wells), while for studies on MIX 2, 50 μL of *S. aureus* PECHA 10 and 50 μL of *P. aeruginosa* PECHA 4 standardized broth cultures were dispensed into each well, and for MIX 3, 33 μL of *S. aureus* PECHA 10, *P. aeruginosa* PECHA 4, and *C. albicans* X3 standardized broth cultures were dispensed. Microtiter plates were incubated for 90 min at 37 °C.

After incubation, the non-adherent cells were gently removed from each well by washing with PBS; adherent cells were treated with 100 μL of L18R peptide at a final concentration of 100 μg/mL (48.35 µmol/L) (or sterile water for the control) and incubated at 37 °C for 5 h for bacteria and 6 h for yeast in the monomicrobial biofilm, and both for 5 h for the polymicrobial biofilm. After incubation, each well was washed with PBS and then fresh TSBG (100 μL) was added for the bacterial biofilm, fresh RPMI-G for the yeast biofilm, and fresh TSBG for the polymicrobial biofilm. The plates were incubated at 37 °C for 24 h.

#### 2.3.2. L18R Treatment on Mature Biofilm

The effect of L18R peptide on mature biofilm (monomicrobial and polymicrobial MIX 2 and MIX 3) was also evaluated. Microbial suspensions were prepared and dispensed as described above. After 24 h of incubation at 37 °C in aerobic conditions, biofilms were treated with 100 μL of L18R peptide at a final concentration of 100 µg/mL (or sterile water for the control) at 37 °C for 5 h for bacteria and 6 h for yeast in the monomicrobial biofilm, and both for 5 h for the polymicrobial biofilm.

#### 2.3.3. Biofilm Biomass Quantification and Determination of CFU/mL

For quantification of biofilm biomass, treated and untreated samples were quantified by 0.25% crystal violet assay (CV, Biolife, Milan, Italy). Briefly, after treatments, wells were washed with PBS, fixed by air drying, stained with 100 μL of CV for 1 min, washed with PBS, and eluted with ethanol. Biofilm biomasses were quantified by measuring absorbance at 540 nm with a microplate reader (BIORAD, Milan, Italy).

For the multispecies biofilm, the number of CFU/mL was also determined. After treatments, wells were washed with PBS, scraped to remove the adhered microorganisms that were re-suspended in 100 μL of PBS, sonicated for 2 min in an ultrasonic bath (Falc, Instrument, Bergamo, Italy), and vortexed for 3 min. Then, 10-fold dilutions were plated on MSA, CET, SAB, and TSA (Trypticase soy agar, TSA, Oxoid, Milan, Italy) and incubated at 37 °C for 24–48 h; then, CFU counting was performed. 

### 2.4. Lubbock Chronic Wound Biofilm Model Assay

The effect of L18R peptide was evaluated on the LCWB model following a previously described methodology [19]. Briefly, bacteria were cultured in TSB and incubated at 37 °C overnight in aerobic conditions, then were standardized to OD_600_ = 0.125 and diluted 1:10 for *S. aureus* PECHA 10 and 1:100 for *P. aeruginosa* PECHA 4, to obtain 10^6^ CFU/mL and 10^5^ CFU/mL, respectively. *Candida albicans* X3 was recovered by SAB and the suspension, prepared in TSB, was standardized to OD_600_ = 0.15 (concentration of ≃5 × 10^5^ CFU/mL).

For the LCWB preparation, 5 mL of Bolton broth (BB, Oxoid, Milan, Italy) with 0.1% agar bacteriological, 50% porcine plasma (Sigma Aldrich, Milan, Italy), 5% horse erythrocytes (BBL, Microbiology System, Milan, Italy), and 2% fetal calf serum (Biolife, Milan, Italy) were dispensed into glass sterile tubes. Subsequently, 5 μL of each diluted broth culture of MIX 3 were inoculated into glass tubes with sterile pipette tips. After 48 h of incubation, the mature biofilms were harvested from the glass tubes, the pipette tips were removed, and the biofilm biomass was washed two times with sterile PBS. After biovolume (V = π × r^2^× h) and biomass weight determination, LCWBs were placed on Petri dishes containing Bolton broth (Oxoid, Milan, Italy) with 1.5% bacteriological agar to produce the “wound bed” for the chronic wound biofilm model [11,19,20].

#### L18R Peptide Treatment on Mature LCWB

After placing the mature biofilm on the “wound bed”, LCWBs were treated with an amount of L18R peptide (at a final concentration of 100 μg/mL), amikacin (AMK, as treatment control, at a final concentration of 64 μg/mL), and PBS (untreated control) with a volume that depended on each LCWB biovolume (V = π × r^2^ × h) [19]. Amikacin was chosen both because it is used in antibiotic treatment for patients with chronic wounds [21] and because it was previously tested in the LCWB model [19,22]. The untreated and treated LCWB biovolumes were compared. The amount was determined in order to avoid the spread of the substances in the Bolton medium wound bed and to permit them to be totally adsorbed only on the LCWB. Next, the treated LCWBs were incubated for 24 h at 37 °C. The biofilm was harvested from the artificial wound bed by using sterile forceps and washed twice with sterile PBS, the excess medium was removed with sterile cotton, and the weight was measured. Subsequently, the biofilm was vortexed for 2 min, sonicated for 3 min (with ultrasound bath), vortexed for another 2 min, and diluted in PBS for the microbial enumeration. The number of CFU/mL was determined by spreading the biofilm on MSA for *S. aureus* PECHA 10, on CET for *P. aeruginosa* PECHA 4, on SAB for *C. albicans* X3, and on TSA for the total count; the plates were incubated at 37 °C for 24–48 h. Data were expressed as CFU/mg of LCWB sample.

### 2.5. Confocal Laser Scanning Microscopy (CLSM) Analysis

The treated and untreated LCWBs were also analyzed by CLSM using live/dead staining BacLight viability kits. The LCWB samples were washed in PBS to selectively remove the non-adhered bacteria and stained with a BacLight kit (Thermo Fisher Scientific, Waltham, MA, USA) for 15 min at room temperature. Samples were then examined using a Zeiss LSM800 microscope (Carl Zeiss, Jena, Germany) coupled to an inverted microscope Axio-observer D1 (Carl Zeiss, Jena, Germany) equipped with a Plan Neofluaroil-immersion objective (100×/1.45 NA). The green and red emission (SYTO 9 and propidium iodide, respectively) were excited using the 488 nm setting (4% of potency) of an argon laser and a helium/neon 543 nm source (2.5% of potency). To separate the fluorescence emissions, HTF 488/543 and NTF 545 as primary and secondary dichroic mirrors, respectively, were used. Detector band-pass filters were set over 505–530 and 565–615 ranges for the green and red emissions, respectively. Images were alternatively recorded using the multitrack acquisition.

### 2.6. Scanning Electron Microscopy (SEM) Analysis

The treated and untreated LCWB samples were fixed with glutaraldehyde, dehydrated with ascending concentrations of ethanol, then immersed in hexamethyldisilazane (HMDS, Sigma-Aldrich, Milan, Italy) for 10 min, twice. HMDS was decanted from the specimen vial and the tissues were left to air dry at room temperature. The dried samples were subjected to gold-sputtering with a desk sputter coater (Phenom-World B.V., The Netherlands) and then observed with SEM (Phenom-World B.V., The Netherlands) at different magnifications, as previously described [23,24].

### 2.7. Statistical Analysis

Each assay was performed at least in triplicate. Data are shown as the means ± standard deviation (SD). Differences between groups were assessed with one-way analysis of variance (ANOVA). *p* values ≤ 0.05 were considered statistically significant.

## 3. Results

### 3.1. Antimicrobial Activity of L18R against Planktonic Bacterial and Fungal Cells

Preliminary experiments were carried out on planktonic bacterial and fungal cells by the colony forming unit (CFU) assay.

L18R exerted a significant antimicrobial activity against all the investigated strains, with half maximal effective concentration (EC_50_) values ranging from 0.278 to 1.157 μM (Table 1). The highest activity was observed against *C. albicans* X3.

### 3.2. Activity of L18R against the Monomicrobial Biofilm

The capability of L18R, at a concentration of 100 μg/mL, to reduce a monomicrobial biofilm formed onto a polystyrene plate was evaluated by microbial biomass quantification through crystal violet (CV) assay.

L18R significantly reduced both early and mature *Candida* biofilm biomass, while *S. aureus* biofilm biomass was less affected (Table 2). A negligible effect was observed on *P. aeruginosa* for the early-stage biofilm, while a slight reduction of the mature biofilm biomass resulted after L18R treatment.

Since L18R treatment reduced *C. albicans* X3 biofilm by more than 90% in both conditions, the EC_50_ value was calculated following treatment with decreasing concentrations of the peptide. The obtained EC_50_ values (and relative 95% confidence intervals) were 0.851 (0.639–1.134) µmol/L and 22.035 (18.215–26.662) µmol/L for early-stage and mature biofilms, respectively.

### 3.3. Activity of L18R against Polymicrobial Biofilm

The activity of L18R against dual species biofilms (*S. aureus* PECHA 10 and *P. aeruginosa* PECHA 4, MIX 2) and on triadic biofilms (*S. aureus* PECHA 10, *P. aeruginosa* PECHA 4, *C. albicans* X3, MIX 3) on polystyrene surfaces was evaluated by CV assay and by CFU determination.

After treatment with L18R, no significant reduction in terms of microbial biomass and CFU/mL was obtained, in comparison with the untreated control, in early-stage and mature biofilms for MIX 2 (not shown).

In the triadic mature biofilm (MIX 3), despite the microbial biomass not being affected in early-stage and mature biofilms, the CFU number was moderately reduced (29%) and a variation of microbial composition was detected (Figure 1) with a slight increase in the amount of *C. albicans*. In all samples, control and L18R-treated, *P. aeruginosa* was the predominant bacterial species followed by *S. aureus* and, in lower numbers, by *C. albicans*.

### 3.4. Activity of L18R in the Lubbock Chronic Wound Biofilm Model

The mature LCWB model including the MIX 3 microbial composition (*S. aureus* PECHA 10, *P. aeruginosa* PECHA 4, *C. albicans* X3) was performed to mimic the in vivo situation.

As shown in Figure 2, after treatment with L18R, a reduction of 30.4% (control, 1.1 × 10^6^ ± 3.8 × 10^5^; L18R-treated, 7.8 × 10^5^ ± 7 × 10^5^) in terms of CFU/mg (total microbial count) was observed. In particular, reductions of 66.6% for *S. aureus* (control, 6.5 × 10^4^ ± 2.7 × 10^4^; L18R-treated, 2.1 × 10^4^ ± 1.6 × 10^4^) and 26.5% for *P. aeruginosa* (control, 1.02 × 10^6^ ± 3.8 × 10^5^; L18R-treated, 7.5 × 10^5^ ± 7 × 10^5^) were detected, whereas the treatment with L18R induced a decrease of 94.4% for *C. albicans* (control, 6.6 × 10 ± 1.3; L18R-treated, 3.7 ± 3, CFU/mg). Treatment with amikacin promoted, in comparison with untreated control, a significant reduction of the total microbial count (80.2%). Nevertheless, the significant decrease of CFU/mg of *P. aeruginosa* (81.1%) was in parallel with a significant increase of *C. albicans* cells.

Confocal laser scanning microscopy (CLSM) and scanning electron microscopy (SEM) analyses were performed to visualize the biofilm structure of both treated and untreated LCWB samples.

The images obtained by CLSM (Figure 3, upper part) displayed an almost exclusive presence of viable cells in the untreated control and a significant increase of dead bacterial cells in the L18R-treated sample. The amikacin-treated sample showed a great number of dead bacterial cells and a remarkable increase in viable yeast cells. SEM images (Figure 3, lower part) confirmed the predominance of *P. aeruginosa* with only a few *Candida* cells in samples treated with L18R, and an evident increase of *Candida* cells in the amikacin-treated sample. The predominant morphology of *Candida* was yeast-like with rare pseudohyphae.

## 4. Discussion

Chronic wound infections represent a crucial factor for the delaying of wound healing; in fact, multispecies and inter-kingdom wound microbial biofilms are notoriously tolerant to drugs and immune clearance, influencing the therapeutic outcome [25,26]. Thus, chronic wounds’ management requires novel solutions for the treatment of resistant microbial colonizers.

Among the innovative approaches recently encouraged in the treatment of resistant microorganisms, AMPs are of particular interest as anti-biofilm agents for their peculiar properties, as outlined by Batoni et al. [20]. In fact, AMPs can target the complex biofilm structure by a double mechanism based on a “classical” microbicidal action, in particular a rapid killing activity, and “non classical” mechanism of action including interference with adhesion and up/down regulation of essential biofilm genes [27]. AMPs can also be easily optimized to increase their effectiveness when used alone or combined with other conventional and unconventional antimicrobial agents [19]. Moreover, peptides of various origins, including defensins, cathelicidins, and histatins, have been reported to be active against fungal biofilms, suggesting their therapeutic potential [27].

L18R is a recently described peptide encoded by the IGHJ2 gene segment that has been proven to be active in vitro against different yeast strains, including *C. albicans* isolates resistant to antifungals. Its therapeutic activity against experimental candidiasis in *Galleria mellonella* has also been demonstrated [16]. L18R showed a rapid candidacidal effect. As demonstrated by confocal microscopy, the peptide bound to the surface of yeast cells in just a few minutes, while progressive internalization and compartmentalization were observed over time, leading to cell death. After the interaction with the cell membrane, different intracellular targets may be involved. Transmission and scanning electron microscopy studies showed morphologic changes in yeast cells after treatment with L18R, such as the presence of microbodies, membrane retraction, and cell wall alterations. Apoptotic cells were also observed, although at a low percentage [16]. L18R showed a random coil conformation in aqueous solution by circular dichroism spectroscopy, but structure–function relationships were not investigated in depth.

In the first step of this study, L18R proved to be effective against planktonic bacterial cells of both Gram negative and Gram positive microorganisms. However, EC_50_ values against bacterial strains, in particular *S. aureus*, were higher than the ones previously reported for yeast strains [16]. Further studies are needed to clarify the mechanism of antibacterial action of L18R. L18R was also able, with different effectiveness, to reduce the biomass of monomicrobial *C. albicans* and *S. aureus* early-stage and mature biofilms. In contrast, against *P. aeruginosa*, no effect was observed in the early-stage biofilm and only a slight effect was seen in the mature biofilm. This is presumably attributable to the production of bacterial exopolysaccharides that are able to interfere with peptide activity and permeability. As reported by Band and Weiss [28], the alginate produced by *P. aeruginosa* in the biofilm extracellular matrix acts as barrier to AMPs diffusion. In fact, anionic alginate could bind cationic AMPs and induce conformational changes, reducing their antimicrobial/antibiofilm action [28].

After treatment with L18R of the polymicrobial biofilm containing *S. aureus* and *P. aeruginosa*, the evaluation of biomass and CFU numbers, both in early-stage and mature biofilms, showed no significant reductions. Again, when the effect of peptide treatment on a triadic biofilm containing *S. aureus*, *P. aeruginosa,* and *C. albicans* was evaluated, no reduction of biomass was observed, neither in early-stage or mature biofilms. Similar results were reported by Townsend et al. [29]. In this study, the authors evaluated the effect of commonly used topical treatments, chlorexidine and povidone iodine, on a 2D polystyrene model in mono and triadic cultures of *S. aureus*, *P. aeruginosa,* and *C. albicans* and on a 3D cellulose matrix in mono and triadic cultures. On the 2D polystyrene monocultures, the two topical agents were able to significantly reduce the bacterial and yeast amounts, but when used in the triadic mixed cultures, the results were different, underlying the complexity of action in the context of microbial multispecies biomass. The lower effectiveness of the topical agents on the in vitro multispecies biofilm model could be related to an increased surface area, different spatial microbial organization, and a different gradient of oxygen and nutrients, which could contribute to microbial tolerance to treatments.

In our study, although L18R was effective against monomicrobial biofilms, it showed only a reduced effect on triadic biofilms on polystyrene surfaces. However, L18R caused a moderate reduction in total CFU number, with a variable effect on different microorganisms. We, therefore, investigated the effect of L18R on mixed bacterial and yeast infections using the LCWB model, which better mimics an in vivo environment [16]. In fact, polystyrene surfaces, although commonly used for studies on mono and multispecies biofilms, are not really representative of the in vivo condition.

As expected, the microbial composition of the LCWB model without any treatment revealed a prevalence of *P. aeruginosa* cells, a lower burden of *S. aureus* cells, and a negligible number of yeast cells. Similarly, Townsed et al., [29], studying the triadic biofilm composed of *S. aureus*, *P. aeruginosa,* and *C. albicans*, showed that the bacterial component represents the most abundant constituent of the total biomass with a clear dominance of *P. aeruginosa*.

Interestingly, when amikacin was used as a control treatment, a significant reduction of *P. aeruginosa* CFUs was observed, along with a smaller reduction of *S. aureus* CFUs, while a significant increase of *C. albicans* CFUs was noticed. In light of this finding, it is clear that a modulating action was exerted by L18R. Peptide treatment caused a reduction, although not a significant one, of total CFUs, mainly due to the decrease of *P. aeruginosa* CFUs and to a lesser extent of *S. aureus* CFUs. The number of *C. albicans* CFUs was clearly decreased compared to that of the control and appeared much lower in comparison to that of the sample treated with amikacin, highlighting the antifungal action of L18R.

These results were also reflected in CLSM and SEM images of LCWB that showed the structure and ultrastructure of the inter-kingdom multispecies biofilm. These images allowed for observation of the spatial distribution of the microbial aggregates and of the predominant bacterial component, which was reduced in the amikacin-treated biofilm where a higher amount of *C. albicans* cells was seen. The presence of *C. albicans* cells in the almost exclusive form of blastoconidia is in agreement with previous reports of triadic biofilms [29]. In particular, *P. aeruginosa* may induce a detrimental effect on the hyphal form of *C. albicans* due to phenazines’ action that could also affect yeast viability.

Despite the fact that the Lubbock model mimics a chronic wound, the limitation of this work is related to the lack of in vivo validation. Further studies will be conducted to confirm our data by using a murine model of wound healing.

## 5. Conclusions

Data obtained in our study underlines the important balancing behavior of L18R in a complex system, including mixed microorganisms and host-derived substances, suggesting its potential role alone or combined with conventional or alternative biocides to target cells embedded in an inter-kingdom biofilm associated with chronic wounds.

Overall, our findings stress the possible employment of antimicrobial peptides as adjunctive therapeutic agents in chronic wound management.

## Figures and Tables

**Figure 1 microorganisms-09-01779-f001:**
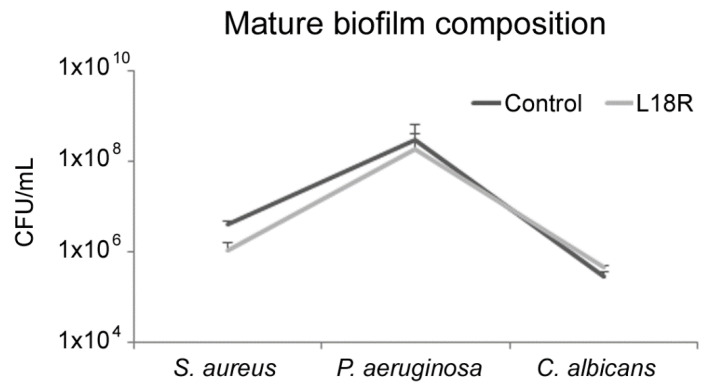
Microbial composition (CFU/mL) of a multispecies mature biofilm (MIX 3) on a polystyrene surface in treated (L18R) and untreated (Control) samples.

**Figure 2 microorganisms-09-01779-f002:**
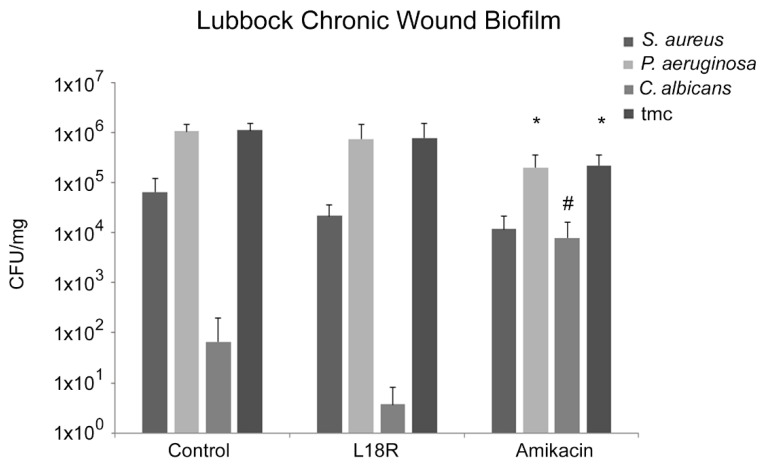
Microbial composition (CFU/mg) of multispecies (MIX 3) in the Lubbock chronic wound mature biofilm after treatment with L18R and amikacin. Black columns represent the total microbial count (tmc). * Statistically significant with respect to the control, # with respect to L18R, (*p* < 0.05).

**Figure 3 microorganisms-09-01779-f003:**
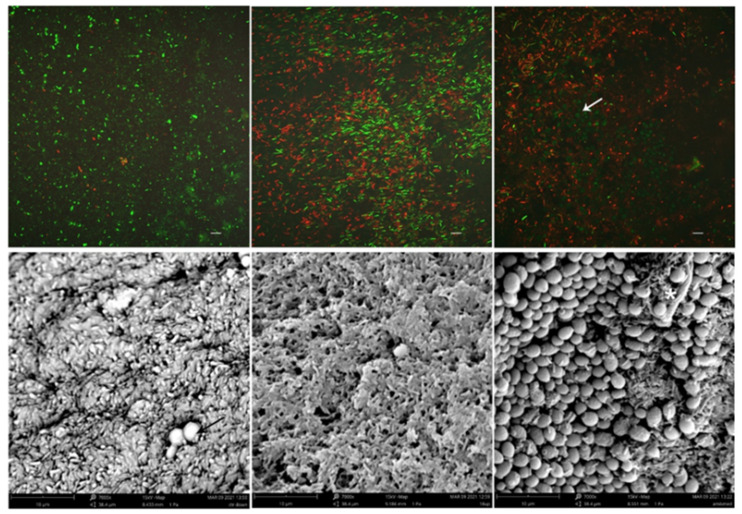
Representative images by CLSM (**upper**) and SEM (**lower**) of LCW mature biofilm. Left panels, untreated control; central panels, L18R-treated samples; right panels, amikacin-treated samples. In control samples, viable bacterial cells (coccoid and rod-shaped green cells in CLSM) are detected, with only a few cells of *C. albicans* (black arrow in the SEM image). After L18R treatment, dead and viable bacterial cells are the main components of the CLSM image. Rod shaped *P. aeruginosa* cells are also predominant on SEM observation. In the amikacin-treated samples, the number of *C. albicans* cells increases with viable yeast cells (white arrow in CLSM image) embedded in a prevalent dead bacterial population and rare pseudohyphae (asterisk) detectable by SEM. CLSM magnification 1.000× (bars = 5 µm), SEM magnification 7.000× (bars = 10 µm).

**Table 1 microorganisms-09-01779-t001:** In vitro activity of L18R against planktonic microorganisms.

Strain	EC_50_ ^1^ (95% Confidence Intervals)
*S. aureus* PECHA 10	1.157 (1.028–1.304)
*P. aeruginosa* PECHA 4	0.372 (0.257–0.539)
*C. albicans* X3	0.278 (0.257–0.300)

^1^ EC_50_, half maximal effective concentration, [mol/L] × 10^−6^.

**Table 2 microorganisms-09-01779-t002:** Reduction of monomicrobial biofilm biomass after treatment with L18R.

Strain	Early-Stage Biofilm	Mature Biofilm
	% Reduction of Biofilm Biomass ^1^	
*S. aureus* PECHA 10	45.65 ± 1.53	57.37 ± 6.45
*P. aeruginosa* PECHA 4	4.75 ± 10.63	20.31 ± 12.98
*C. albicans* X3	97.19 ± 1.02	98.81 ± 1.68

^1^ % reduction evaluated by CV assay with reference to untreated control.

## Data Availability

The data supporting the findings of this study are available within the article.

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
