# Peer review of "Antimicrobial Peptide L18R Displays a Modulating Action against Inter-Kingdom Biofilms in the Lubbock Chronic Wound Biofilm Model"

_microorganisms, 2021, doi:10.3390/microorganisms9081779_

Round 1

Reviewer 1 Report

The work entitled “Antimicrobial peptide L18R displays a modulating action against inter-kingdom biofilm in Lubbock Chronic Wound Biofilm model” by Fermo et al. investigates the effects of the antifungal peptide L18R against three representative wound pathogens: Staphylococcus aureus, Pseudomonas aeruginosa, and Candida albicans. The work is well done and well organized. The acquired data is scientifically sound and is discussed in light of recent research on the field. Overall, the work is of great impact in this area and should be considered for publication after minor revision:

  • English writing must be fixed – small mistakes are detectable
  • Add the section conclusions to the manuscript.

Author Response

The work entitled “Antimicrobial peptide L18R displays a modulating action against inter-kingdom biofilm in Lubbock Chronic Wound Biofilm model” by Fermo et al. investigates the effects of the antifungal peptide L18R against three representative wound pathogens: Staphylococcus aureus, Pseudomonas aeruginosa, and Candida albicans. The work is well done and well organized. The acquired data is scientifically sound and is discussed in light of recent research on the field. Overall, the work is of great impact in this area and should be considered for publication after minor revision:

  • English writing must be fixed – small mistakes are detectable
  • Add the section conclusions to the manuscript.

Answer

The small mistakes were corrected in all part of the Manuscript and the conclusion section was improved (pag. 9, lines 34-42)

The conclusions section has been included

Reviewer 2 Report

A very interesting and complex work

Microorganism in biofilms with polymicrobial composition can be highly resistant to treatment, so the idea to study the effects of new compounds on complex biofilm using models that mimic the physiological environment is very good.

I have a question about the use of amikacin as a control treatment. This was not mentioned in the introduction. Is it a standard protocol when testing be biofilm complex activity in the LCWB model? or was amikacin chosen for another reason? Please clarify.

Author Response

A very interesting and complex work

Microorganism in biofilms with polymicrobial composition can be highly resistant to treatment, so the idea to study the effects of new compounds on complex biofilm using models that mimic the physiological environment is very good.

I have a question about the use of amikacin as a control treatment. This was not mentioned in the introduction. Is it a standard protocol when testing be biofilm complex activity in the LCWB model? or was amikacin chosen for another reason? Please clarify.

Answer

The use of Amikacin as control has been clarified in the Materials and Methods section also including 2 new references (20, 21, new version of the MS)

Reviewer 3 Report

Reviewed article is good written. Authors used biofilms obtained from single strains and additionally Lubbock Chronic Wound Biofilm model containing three pathogens S. aureus, P. aeruginosa and C. albicans. Methodology and results are properly presented. I would like only suggest add more information about L18R, its structure and/or 3D model.

Author Response

Reviewed article is good written. Authors used biofilms obtained from single strains and additionally Lubbock Chronic Wound Biofilm model containing three pathogens S. aureus, P. aeruginosa and C. albicans. Methodology and results are properly presented. I would like only suggest add more information about L18R, its structure and/or 3D model.

Answer

A sentence has added in the Discussion section. (Pag 9 lines 19-28)

Reviewer 4 Report

The authors investigated the modulating activity of antimicrobial peptide L18R on an inter-kingdom biofilm in a biofilm model. The fact that antimicrobial compounds modulate biofilm communities is known for decades but the authors failed to tell us what their working hypothesis is and what they wanted to achieve with their experiments. In general, this manuscript is lacking many explanations and justifications.

Some sentences of explanation for the model used would be highly welcomed. What are the advantages against other infection models, e.g. Galleria mellonella, C. elegans, etc.? What are the limitations?

Why L18R and not any of the other many known AMPs is absolutely not clear. We know for several AMPs that they act selectively on specific bacteria, therefore these results are not surprising. But why do they act selectively? The authors did not try to shed some light on this open question.

The authors remained very vague what the significance of their study is. All I could spot is that L18R showed some selectivity but this is almost trivial because as stated above this is known for several AMPs and antibiotics at sub-toxic concentrations. The authors did not offer a way how this could help in medical applications.

How were the isolates identified? At least 16S rRNA gene sequencing is required and here the sequences have to be deposited in a public databank and the accession numbers have to be given in the text for easy access.

Author Response

The authors investigated the modulating activity of antimicrobial peptide L18R on an inter-kingdom biofilm in a biofilm model. The fact that antimicrobial compounds modulate biofilm communities is known for decades but the authors failed to tell us what their working hypothesis is and what they wanted to achieve with their experiments. In general, this manuscript is lacking many explanations and justifications.

Some sentences of explanation for the model used would be highly welcomed. What are the advantages against other infection models, e.g. Galleria mellonella, C. elegans, etc.? What are the limitations?  are the limitations?

Answer

As reported in the introduction section “The LCWB is in vitro chronic wound system for multispecies consortium studies” and it mimics the in vivo microbial distribution in a chronic wound. It is important to recognize that this approach is an in vitro model and not applicable to the study of dynamic biofilm–host interactions in vivo. The Lubbock model may be useful to test the efficacy of antimicrobial agents against biofilm versus planktonic microbes.

In the new version of the Manuscript was inserted a new sentence regarding the limitation of the study (Pag 9, lines 32-34)

Moreover, We better specify the sentence in the Introduction, pag 2 lines 31-32. “ In addition, L18R proved to be effective in an experimental model of systemic infection by C. albicans in larvae of Galleria mellonella, displaying no hemolytic, cytotoxic, or genotoxic activity in mammalian cells.”

Why L18R and not any of the other many known AMPs is absolutely not clear. We know for several AMPs that they act selectively on specific bacteria, therefore these results are not surprising. But why do they act selectively? The authors did not try to shed some light on this open question.

The authors remained very vague what the significance of their study is. All I could spot is that L18R showed some selectivity but this is almost trivial because as stated above this is known for several AMPs and antibiotics at sub-toxic concentrations. The authors did not offer a way how this could help in medical applications.

Answer

We choose L18R, a peptide recently described by some of the coauthors (Ref. 15 Polonelli, L.; Ciociola, T.; Sperinde, M.; Giovati, L.; D'Adda, T.; Galati, S.; Travassos, L.R.; Magliani, W.; Conti, S. Fungicidal activity of peptides encoded by immunoglobulin genes. Sci Rep 2017, 7, 10896. doi: 10.1038/s41598-017-11396-6) that showed a good candidacidal activity, to evaluate its potential also against bacterial cells.  While some hypotheses were made with regard to L18R mechanism of action against Candida albicans, further studies are needed to elucidate L18R activity against bacteria.

How were the isolates identified? At least 16S rRNA gene sequencing is required and here the sequences have to be deposited in a public databank and the accession numbers have to be given in the text for easy access.

Answer

In the new version of the MS, we included an other our reference (ref 18) in which we used some isolates. We also specify the ethic Committee approval  number (ID n. richycnvw, Inter Institutional Ethic Committee of University “G. d’Annunzio” Chieti-Pescara, Chieti, Italy.

Reviewer 5 Report

1.Over all the research very  intresting and well organised, but present paper found more typographical and syntx errors,further  english improvement needed

2.what  is the biofilm  biovolume and thickness of the  control and treatment biofilm,author should be mention the manuscript

2.Separate the conclusion part and add the future studies

Author Response

1.Over all the research very intresting and well organised, but present paper found more typographical and syntxerrors, further  english improvement needed

2.what is the biofilm  biovolume and thickness of the control and treatment biofilm,author should be mention the manuscript

2.Separate the conclusion part and add the future studies

Answer

The small mistakes were corrected in all part of the Manuscript and the conclusion section was inserted (pag. 9, lines 35-41)

A new sentence was inserted (Pag 4 lines 31-32) to better specify the comparison among biovolumes

Round 2

Reviewer 4 Report

The authors did only a minimalistic revision of their manuscript which is in some places not enough.

Some sentences on AMPs, their diversity, function, mode of action would have boosted the understanding of the background for many readers. Maybe a citation of a good and recent review in this filed would help as well.

The way how the strains have been identified is still missing. The reference is not enough here.

There is no such thing as anonymized clinical strains. You probably mean strains from anonymized patients?

Author Response

The authors did only a minimalistic revision of their manuscript which is in some places not enough.

Some sentences on AMPs, their diversity, function, mode of action would have boosted the understanding of the background for many readers. Maybe a citation of a good and recent review in this filed would help as well.

The way how the strains have been identified is still missing. The reference is not enough here.

There is no such thing as anonymized clinical strains. You probably mean strains from anonymized patients?

Answer

Thanks for the useful suggestions
In the new version of the MS:

- we included a new reference on AMPs ( Kumar, P.; Kizhakkedathu, J.N.; Straus, S.K. Antimicrobial Peptides: Diversity, Mechanism of Action and Strategies to Improve the Activity and Biocompatibility In Vivo. Biomolecules. 2018, 8, doi:10.3390/biom8010004)

-we modified the sentence as the referee suggests “Clinical strains from anonymized patients”…pag 2, line 43

-we better explain how the strains have been identified:

“Clinical strains were isolated from chronic wounds swab culturing onto Mannitol Salt Agar (MSA, Oxoid, Milan, Italy), Cetrimide agar (CET, Oxoid, Milan, Italy) and Sabouraud agar medium (SAB, Oxoid, Milan, Italy). All plates except SAB plates were incubated aerobically at 37°C for 24 h. SAB plates were also incubated aerobically at 30°C for 48 h. Microorgnisms were identified by morphologic aspects of the colonies, followed by biochemical identification using the automated Vitek 2 system (bioMerieux, Marcy ´ l’Etoile, France)” (Pag 3, lines 46-52)